# A Useful Method for the Practice of Pneumatic Retinopexy: Slit-Lamp Laser Photocoagulation through the Gas Bubble

**DOI:** 10.3390/jpm13050741

**Published:** 2023-04-27

**Authors:** Aslan Aykut, Mehmet Orkun Sevik, Betül Kubat, Volkan Dericioğlu, Özlem Şahin

**Affiliations:** Department of Ophthalmology, Marmara University School of Medicine, Istanbul 34854, Turkeykubatbetul@gmail.com (B.K.); volkandr@gmail.com (V.D.)

**Keywords:** laser retinopexy, pneumatic retinopexy, retinal detachment, sulfur hexafluoride, wide-field contact lens

## Abstract

This study aimed to demonstrate the laser retinopexy method through the gas bubble under a slit-lamp biomicroscope using a wide-field contact lens to treat rhegmatogenous retinal detachment (RRD) with pneumatic retinopexy (PR) and report its anatomical and functional results. This single-center, retrospective case series included RRD patients treated with PR using sulfur hexafluoride (SF_6_). The demographics, preoperative factors, and anatomical and functional outcomes were collected from the patient files. The single-procedure success rate of PR at postoperative 6th months was 70.8% (17/24 eyes), and the final success rate after secondary surgeries was 100%. The BCVA was better in the successful PR eyes at postoperative 3rd (*p* = 0.011) and 6th month (*p* = 0.016) than in failed eyes. No single preoperative factor was associated with PR success. The single-procedure success rate of PR using the laser retinopexy method through the gas bubble with a wide-field contact lens system seems comparable to the PR literature.

## 1. Introduction

Rhegmatogenous retinal detachment (RRD) is a commonly occurring type of retinal detachment. This condition arises due to a break in the neurosensory retina, allowing fluid from the vitreous cavity to seep into the subretinal space, resulting in the separation of the neurosensory retina from the retinal pigment epithelium. About 1 in 10,000 people are estimated to be affected by RRD each year [1]. RRD typically affects individuals over 50 years old, with men being more susceptible than women. In addition, certain risk factors such as severe myopia, previous ophthalmic surgery, or ocular trauma can also increase the likelihood of developing RRD [1].

It has been exactly a century since the idea emerged that RRD could be treated by intraocular gas injection [2]. The PR method involves the injection of a gas bubble into the vitreous cavity of the eye to push the detached retina back into place. However, the most critical part of the treatment, the formation of the chorioretinal adhesions, could not be managed solely by this technique introduced. Therefore, it took more than 70 years to describe the pneumatic retinopexy (PR) method accepted in the modern world [3,4]. Several randomized controlled trials have since demonstrated the success of PR in the treatment of RRD. For instance, a study by Tornambe and Hilton [5] reported a success rate of 86% with PR, compared to 69% with scleral buckling (SB) surgery. Another study by Hilton and Grizzard [6] showed that PR had a success rate of 82%, compared to 84% with SB. Moreover, a meta-analysis by Sena et al. [5] found that PR had a higher success rate (87.6%) than SB (80.2%). While PR has shown a higher success rate than SB in some studies, the treatment choice may depend on several factors, such as the location, the presence of proliferative vitreoretinopathy (PVR), the extent of the retinal detachment, and the surgeon’s preference and experience [6].

Although PR has been shown to be as effective as SB in terms of anatomic success rates, PR is associated with better visual outcomes and fewer complications [6,7,8]. For example, a study by Hilton and Grizzard [6] reported that PR had a lower incidence of diplopia (3% vs. 15%) and induced astigmatism (4% vs. 11%) than SB. Similarly, Sena et al. [5] found that PR had a lower risk of induced myopia, infection, and choroidal detachment compared to SB. Randomized controlled trials subsequently demonstrated the success of PR in the RRD [7,8]. Studies comparing SB and PR showed similar anatomic success rates with morbidity and visual outcomes favoring PR [5,8,9]. The choice of retinopexy in those studies was either laser photocoagulation or cryotherapy; however, the difference in success rates between laser and cryotherapy was not evaluated [5,8,9].

Since its inception, pars plana vitrectomy (PPV) has become the most preferred surgical method for treating RRDs in the world [10,11,12]. A prospective, randomized-controlled, multicenter study comparing SB and PPV reported a single-procedure success rate of 63.6% and 63.8% and a final anatomical success rate of 96.6% and 96.7% for SB and PPV, respectively [13]. PR and PPV for RRDs were recently compared in a prospective, randomized-controlled, multicenter study [14]. The single-procedure success rates at 12 months were 80.8% and 93.2%, and final anatomical success rates were 98.7% and 98.6% for PR and PPV, respectively. One of the limitations of this study was the inhomogenous retinopexy method used in the PR group, including cryotherapy alone, laser retinopexy alone, or a combination of both. Therefore, the effects of the retinopexy techniques could not be evaluated [14].

Laser retinopexy is one of the surgical techniques used to treat retinal detachment. It can be performed with an indirect ophthalmoscope using a non-contact lens or under a slip-lamp biomicroscope through contact lenses. Compared to cryotherapy, laser retinopexy has several advantages. It creates faster and more stable chorioretinal adhesions, which can help prevent the retinal detachment from progressing. The findings of previous studies have demonstrated that laser photocoagulation effectively produces strong bonds between the neurosensory retina and the retina pigment epithelium and the choroid, almost equivalent to the normal bond strength within a day after the procedure [15]. The cause behind this rapid bonding is speculated to be the formation of fibrin within the treated area. As time passes, the bond between the structures becomes even more robust, with its strength reaching approximately twice the normal level after 2–3 weeks [16]. Cryotherapy, on the other hand, appears to result in weaker adhesion during the initial week following the procedure, possibly due to inflammation or edema in the surrounding tissue. However, the adhesion gradually strengthens over the following week, reaching a level comparable to laser retinopexy [15]. In addition, laser retinopexy is less invasive than cryotherapy and does not require extremely cold temperatures. This can make it a more comfortable and convenient option for patients and has an advantage over cryotherapy in producing faster and more stable chorioretinal adhesions [17,18]. Cryopexy is also likely a stimulating factor for postoperative PVR in retinal detachments due to horseshoe tears with involuted trailing edges or retinal tears of 180 degrees and larger [19]. Another risk with cryopexy is the formation of large areas of scarring from the cryopexy field, which can lead to the development of rhegmatogenous retinal detachment years after treatment [20].

Even well-designed studies in the literature have focused mainly on patient selection rather than retinopexy methods [8,14]. However, using multiple retinopexy methods might lead to different outcomes in the early and late phases. Therefore, there is still a potential need for retinopexy techniques and their effectiveness to be investigated further. The purpose of this study is to point the laser retinopexy method through the gas bubble under a slit-lamp biomicroscope with a wide-field contact lens after intravitreal expansile gas injection for RRD treatment and report its anatomical and functional results.

## 2. Materials and Methods

This single-center, retrospective case series included all patients presenting as primary RRD and were treated with PR between January 2018 and December 2021 in Marmara University School of Medicine, Department of Ophthalmology, Vitreoretinal Surgery Unit. The study protocol was approved by the Institutional Review Board of Marmara University School of Medicine (No: 09.2021.371) and followed the tenets of the Declaration of Helsinki. In addition, all patients provided written informed consent for the treatment and had their medical information used in the study analysis.

### 2.1. Pneumatic Retinopexy

Detachments with all retinal tears located between superior 8- and 4-clock hours positions are evaluated for PR, and more than two breaks are allowed for PR even if they are located more than one clock hour apart. Patients with inferior retinal tears located between 8 and 4 clock hours, retinal dialysis, giant retinal tears (more than 3 clock hours), PVR grade C or worse according to revised 1991 criteria [21], significant media opacity that would prevent external retinal photocoagulation, inability to tolerate wide-field contact lens examination, and inability to posture postoperatively were excluded from PR protocol.

All PR procedures were performed by the same surgeon (AA) under topical anesthesia in a sterile operating room within 24 h of the patient’s admission. Any surgical procedure such as cryopexy or laser retinopexy for lattice degeneration before the PR procedure was not applied. Forty-five minutes before the procedure, intraocular pressure decompression is provided by topical 2% dorzolamide/0.5% timolol combination and 0.15% brimonidine, oral 500 mg acetazolamide, and intravenous 20% mannitol 5 mL/kg. During the procedure, at first, an anterior chamber paracentesis was created with a 20 G microvitreoretinal (MVR) blade for additional decompression of the globe. Then, 0.6 mL of 100% sulfur hexafluoride (SF_6_; Teknomek, Istanbul, Turkey) in a 2 mL injector is injected intravitreally with a 26 G needle. The patient’s light perception is immediately checked after the intravitreal gas injection to assess if the perfusion of the optic nerve and retina is compromised. Additional drainage from the paracentesis is allowed if the patient has no light perception until the recheck reveals the perception of light. Finally, the patient is removed from the operating room in the face-down position.

The patients were hospitalized overnight after the intravitreal gas injection in the face-down position, no steamroller maneuver was employed in any patient, and all patients were reevaluated within 24 h after the injection. Laser retinopexy was initiated immediately in the consequent evaluation if the retina is attached around the break, even if the subretinal fluid persists in any other quadrants. All laser retinopexy procedures were performed through the gas bubble by the same surgeon (AA) with Visulas 532 s device (Carl Zeiss Meditec, Jena, Germany) using Volk SuperQuad^®^ 160 (Volk Optical Inc. Mentor, OH, USA) wide-field contact lens system. Laser parameters were adjusted to obtain opaque off-white burns as 100 µm spot size, 300 to 600 mW power, and 0.1 s duration. Prophylactic 360-degree laser photocoagulation was not performed in any patient. An additional laser retinopexy session was conducted if the laser could not be completed in a single session due to corneal epithelial edema or patient discomfort. Detachment in any quadrants within 48 h after injection, even if the edges of the tear were lasered, was considered a failure. In any case of failure, all patients were referred to PPV without any additional gas injection.

### 2.2. Patient Data

Retrospective data of patients collected from the patient files were on demographics (age, gender), preoperative and postoperative (1, 3, and 6 months after the last operation), best-corrected visual acuity (BCVA), lens status (phakic or pseudophakic), the time between symptoms and the presentation (days), count of retinal tears, area of the detachment (clock hours), macular status (macula on (attached) or off (detached)), PVR grade according to revised 1991 criteria [21], follow-up time (months), and if any failure occurred; the reason of the failure, the time of failure and secondary surgeries applied during the follow-up.

### 2.3. Statistical Analysis

Statistical data analysis was conducted by Statistical Package for the Social Sciences (SPSS) for Windows version 22.0 (IBM Corp., Armonk, NY, USA). The descriptive data were presented as mean ± standard deviation (median; minimum–maximum), and qualitative variables were given as frequency (n) and percentage (%). For statistical analysis, the BCVA evaluated with an electronic Snellen chart was converted to the logarithm of the minimum angle of resolution (logMAR) values. The logMAR equivalents of “counting fingers” and “hand motion” visual acuities were accepted to be 1.85 and 2.30, respectively [22]. Qualitative data were compared with Pearson Chi-square or Fisher’s Exact tests. Preoperative and postoperative BCVA were compared by Friedman test and pairwise comparisons with Bonferroni post-hoc correction for multiple comparisons. In addition, independent samples were compared with the Mann–Whitney U test. A *p*-value of <0.05 was considered statistically significant, and Bonferroni-adjusted *p*-values were given where appropriate.

## 3. Results

Twenty-four eyes of 23 RRD patients (12 female (52.2%); 11 male (47.8%)) treated with PR were included in the study analysis. The mean age of the patients was 59.1 ± 8.9 (58; 43–74), and 18 (75.0%) of the eyes were phakic.

Most patients (79.2%) presented within ten days of visual loss or visual field loss with a mean symptom duration of 8.7 ± 9.2 (6.0; 1–30) days. At presentation, 14 eyes (58.3%) had macula on RRD, and the mean detached retinal area was 3.9 ± 1.6 (4; 1–7) clock hours. Twenty patients (83.3%) had PVR grade A, and four had grade B (16.7%). All retinal tears were between superior 4–8 clock hours with a mean count of 1.3 ± 0.6 (1; 1–3), and lattice degeneration was present in 6 (25%) eyes.

Seventeen of twenty-four eyes achieved anatomical success after PR, establishing a single-procedure success rate of 70.8%. All patients with failed PR had PPV with either 1000 cst silicone oil (Teknomek, Istanbul, Turkey) tamponade (6 eyes; 85.7%) or 14.0% perfluoropropane (C_3_F_8_, Teknomek, Istanbul, Turkey) gas tamponade (1 eye; 14.3%). All eyes with silicone oil had a silicone oil removal surgery within six months. An anatomical success rate of 100% was achieved in all eyes six months after their last surgery.

The reasons for the failure of PR were a new or missed retinal break in 5 patients (71.4%) and nonclosure of the initial retinal tear in 2 patients (28.6%). In univariate analysis, none of the evaluated preoperative factors; sex, lenticular status, the time between symptoms and presentation, retinal tear counts, the extent of retinal detachment, macular status, and PVR grade, were associated with the outcome of PR (Table 1).

The mean BCVA of the whole cohort at the presentation was 0.70 ± 0.70 (0.46; 0.00–2.30) logMAR. Although the mean BCVA improved to 0.41 ± 0.30 (0.30; 0.00–1.00) at the postoperative 1st month, to 0.37 ± 0.28 (0.30; 0.00–1.00) at the postoperative 3rd month, and to 0.36 ± 0.33 (0.30; 0.00–1.30) at the postoperative 6th month, the differences were not statistically significant, (χ^2^[3] = 6.156, *p* = 0.104). When the eyes with a successful single PR procedure were evaluated, BCVA was statistically significantly changed from preoperative through postoperative six months (χ^2^[3] = 17.806, *p* < 0.001). Post hoc analysis revealed statistically significant differences from preoperative (0.78 ± 0.75 (0.52; 0.00–2.30) to postoperative 3rd (0.28 ± 0.25 (0.22; 0.00–0.70), *p* = 0.047) and to 6th months (0.26 ± 0.26 (0.22; 0.00–0.70), *p* = 0.017), but not to 1st month (0.32 ± 0.27 (0.30; 0.00–0.80), *p* = 1.000). There was also no significant change in BCVA from 1st to 3rd (*p* = 0.978), 1st to 6th (*p* = 0.505), and 3rd to 6th months (*p* = 1.000). Considering the eyes that failed in PR but achieved anatomical success after PPV, the preoperative BCVA (0.52 ± 0.56 (0.30; 0.00–1.30)) was not statistically different at 1 month (0.61 ± 0.31 (0.52; 0.22–1.00)), 3 months (0.61 ± 0.22 (0.52; 0.30–1.00)) and 6 months (0.62 ± 0.36 (0.52; 0.22–1.30)) after the last surgery (χ^2^[3] = 2.262, *p* = 0.520). Although the BCVA was not statistically different between the eyes with successful single PR procedure and failed eyes preoperatively (*p* = 0.494) and postoperative 1st month (*p* = 0.055), it was significantly better in the eyes with successful single PR procedure at postoperative 3rd (*p* = 0.011) and 6th months (*p* = 0.016) (Figure 1).

## 4. Discussion

In this single-center retrospective case series, a single-procedure anatomical success rate of 70.8% at the postoperative 6th month was observed with PR protocol using the laser retinopexy method through the gas bubble under a slit-lamp biomicroscope with a wide-field contact lens system. Although our success rate seems lower than the randomized controlled trials conducted (70–88%) [8,14,23,24], it is comparable to real-life retrospective studies (61–78.2%) [25,26,27,28,29,30,31].

Lower success rates in real-life studies might result from more stringent inclusion criteria of randomized trials, particularly considering PVR grades, retinal tear counts, and clock hour areas involving retinal tears [32]. In the multicenter randomized trial comparing PR and SB with a single PR success rate of 73% by Tornambe et al. [8], as well as The pneumatic retinopexy versus vitrectomy for the management of primary rhegmatogenous retinal detachment outcomes randomized trial (PIVOT) [14] with a primary anatomic success rate of 80.8%, one of the main inclusion criteria was a single retinal tear or multiple tears located within one clock hour area of retinal detachment. However, in our protocol, all retinal tears are allowed for PR even if they are located more than one clock hour apart as long as they are located between the superior 4–8 clock hours. Moreover, PVR grades of B or worse were not recruited in the PIVOT trial; however, in the study of Tornambe et al. [8], eyes with a PVR grade of B were allowed for PR like our study, which might explain our closer single-procedure success rates to Tornambe et al. [8] (i.e., 73% vs. 70.8%). In addition, the final anatomical result with secondary procedures in our study (100%) was also comparable with Tornambe et al. [8] and the PIVOT trial [14], which were 99% and 98.7%, respectively. In a single-center retrospective study, traditional PR, including the eyes with the recruitment criteria of Tornambe et al. [8], and nontraditional PR, defined as eyes with characteristics that might adversely affect procedure outcomes such as more PVR, giant or inferior retinal tears, more than one retinal tear separated by more than one clock hour, etc., were compared [28]. Although it was not statistically significant (*p* = 0.16), the study resulted in lower single-procedure success rates with nontraditional PR (74.4%) than with traditional PR (84.1%), which was closer to our results, indicating lower success rates with expanded PR criteria in real-life [28].

The most common complications of gas injection are subconjunctival gas, cataract development, subretinal gas, and fish-egg formation [33]. Fish egg formation, in particular, is the most important complication that complicates the retinopexy phase. No fish egg formation was observed in our study. It has been said in the literature that the best injection method is to use a tuberculin syringe and a 27–30 gauge needle [17,34]. However, in our prior cases (unpublished data), we observed the formation of fish eggs when using this technique. As a result, we chose to use a 2 mL syringe and 26 G injector in this series, believing that these tools would deliver the gas more rapidly and uniformly, as detailed in the methodology section.

The management of RRDs differs between the continents and even within the different regions of the same country [11,35]. PPV is the most used method in Europe and the USA; however, PR and SB are preferred less in Europe [10,11,12]. Cost analyses indicate pneumatic retinopexy to be more cost-effective than SB and PPV, with savings as high as 50.9% compared with SB and 59.4% compared with PPV [28]. In addition, considering the quality of life and financial gains, it is noted that PR is more advantageous than SB and PPV [36]. A 2013 survey by the American Society of Retinal Specialists showed that only 25 percent of surgeons would prefer this technique to PPV in cases where pneumatic retinopexy is indicated. Moreover, this low rate is below the survey conducted about ten years ago. There appears to be a clear regression in favor of pneumatic retinopexy [37]. There may be some reasons for this. One of the reasons may be that modern PR is a relatively newer method than SB; however, SB, a much older method, is also being preferred at a decreasing rate [38]. It was suggested that the lack of training and experience or insufficient financial repayment might cause the unpopularity of those techniques [38].

In one study evaluating the learning curve in the SB procedure, the authors noted a relationship between SB success and experience; however, whether the ability to use indirect ophthalmoscopy, which is an essential part of this method, affected the results was not stated [38]. The inadequacy of indirect ophthalmoscopy at the cardinal steps of SB was noted in one study comparing SB with wide-angle viewing systems (WAVS) and conventional SB [39]. Similarly, a more recent study reported that SB and PPV had similar success rates when wide-angle viewing systems (WAVS) were used in both methods in the treatment of RRDs [40]. They also stated that the single-procedure success rate of SB with this viewing system is higher than in previous studies with the traditional methods, emphasizing the inadequacy of indirect ophthalmoscopy during retinal tear identification and retinopexy stages [40]. To our knowledge, no studies evaluated the success of PR in the context of visualization systems. However, since there is evidence that indirect ophthalmoscopy is not the best visualization method for SB, one could argue that a similar idea might apply to PR.

Considering the previously mentioned randomized or real-life studies investigating PR in the literature, it should be noted that the success rates were assessed and compared regarding the eyes’ suitability for PR procedure without taking into account the retinopexy stage [8,14,23,24,25,26,27,28,29,30,31]. Moreover, a recent big data study involving approximately 10,000 patients using the IRIS system in the USA even found that gender and smoking might be associated with a single-procedure success rate of PR for the first time [30]. However, there was a lack of data to analyze the retinopexy method [30]. Likewise, other big data studies conducted on Medicare provided information about vitreoretinal trends for RRD; however, there were no data on retinopexy trends since there was no separate billing for the retinopexy method [41].

Today, ophthalmologists frequently use slit-lamp biomicroscope imaging to evaluate and treat posterior segment pathologies [42]. Additionally, since modern retinal photocoagulation devices are developed on the slit lamp biomicroscopes and retina specialists are more familiar with the laser on this platform, experience gained from other laser photocoagulation processes can be transferred to [43]. Therefore, the laser retinopexy method through the gas bubble under a slit-lamp biomicroscope with a wide-field contact lens can be considered a valid approach during the PR procedure.

Among the real-life PR studies in the literature, the vitreoretinal fellows from six training centers across the USA performed one of the lowest single-procedure success rates (66.8%) [27]. In the study mentioned above by Emami-Naeini et al. [27], although there was no significant difference noted between the vitreoretinal fellows in the first and second years of training (61.1% and 68.2%, respectively), the authors noticed a trend towards greater anatomical success rates regarding the eyes treated within the fellows’ first 15 cases versus within 16 or more cases (60.3–63.2% vs. 86.2%, respectively, *p* = 0.08). Considering that the indications remained constant in the training centers, the increasing trend in success rates indicates the importance of experience in PR. Furthermore, in our series, although all stages of the PR procedure were performed by a single surgeon who had previously applied the technique, the success rate also, although not statistically significant, had a higher trend in the second half of the cases, additionally supporting the importance of the experience or keep practicing the method.

Although there is a common approach to the gas delivery technique, a gold-standard retinopexy method has yet to emerge in the literature. The most commonly used techniques for retinopexy in the literature are cryopexy [8,14,25,26,28] and laser indirect ophthalmoscope [8,14,25,28] or less commonly slit-lamp [24], indirect laser retinopexy performed after moving the gas bubble away from the treatment area. According to the literature, a laser indirect ophthalmoscope is considered to be a more effective method than slit-lamp delivery of laser in bypassing the hindrance caused by intraocular gas bubbles, particularly when accessing retinal breaks on the peripheral fundus [18,44]. Furthermore, it is recommended to be cautious while treating through intraocular gas bubbles and avoid using excessive laser power since these bubbles can insulate the heat generated by the laser. In fact, it is recommended to avoid using the laser through the gas bubbles altogether, as the aberrant optical effects produced by the bubbles can lead to inaccurate or harmful laser application [33]. However, our method involves performing laser through gas, which is different from what is suggested in the literature. However, we did not observe any new retinal tears in any case due to incorrect laser application or heat increase. To the best of our knowledge, there are no studies comparing the success rates of PR according to retinopexy methods. Although cryopexy is a technically more straightforward method, it has some disadvantages, such as disrupting the blood–retina barrier, causing retinal pigment epithelial dispersion more than laser photocoagulation, and subsequently causing more frequent development of PVR [45,46].

While considering the results of this study, several limitations must be mentioned. First, the number of eyes involved in the study is relatively small compared to other PR studies [8,14,25,26,27,28,29]. This limitation also prevents us from commenting on the indications for PR, especially regarding the PVR stage and the clock hours distance between the retinal tears. In addition, the laser retinopexy method is not compared to other previously mentioned methods; therefore, it can not be suggested as a more appropriate method. However, the single operation success rate of PR using the laser retinopexy method through a gas bubble with a wide-field contact lens system is comparable to previous studies. Therefore, this study may lead to further randomized studies comparing the success of different retinopexy methods used in the PR procedure.

## Figures and Tables

**Figure 1 jpm-13-00741-f001:**
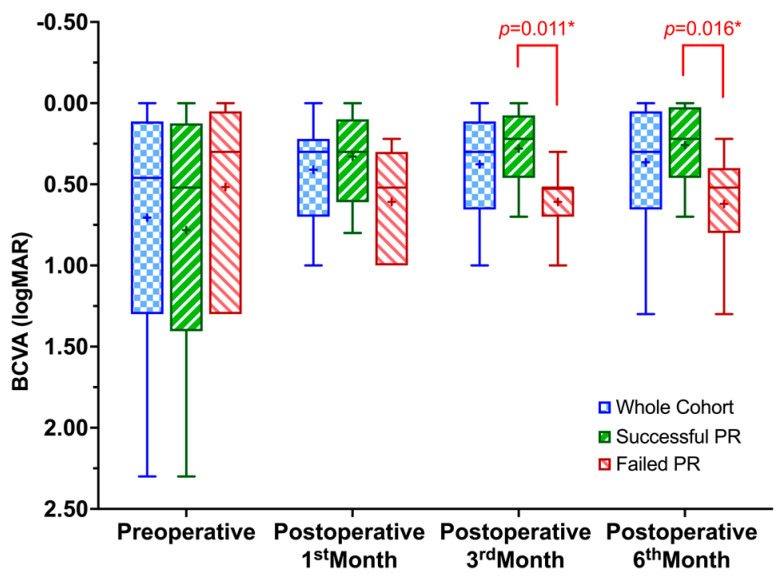
Box and whiskers plot showing the change in best-corrected visual acuity in the whole cohort and the eyes with successful or failed pneumatic retinopexy procedure. BCVA, best-corrected visual acuity; logMAR, logarithm of the minimum angle of resolution; PR, pneumatic retinopexy. The “+” sign indicates the mean values. * Mann-Whitney U Test.

**Table 1 jpm-13-00741-t001:** Univariate analysis of preoperative factors might be associated with the pneumatic retinopexy results.

Preoperative Factors	Pneumatic Retinopexy Result		*p*
	Success	Failure	
**Sex**, *n* (%) Male Female			0.605 *
	8 (47.1)	3 (42.9)	
	9 (52.9)	4 (57.1)	
**Lens Status**, *n* (%) Pseudophakic Phakic			0.586 *
	4 (23.5)	2 (28.6)	
	13 (76.5)	5 (71.4)	
**The time between symptoms and presentation**, days Mean ± SD Median (min–max)			0.757 ^†^
	10.2 ± 10.7	5.3 (2.4)	
	5 (1–30)	7 (1–7)	
**Number of retinal tears**, *n* Mean ± SD Median (min–max)			0.418 ^†^
	1.2 ± 0.4	1.6 ± 0.8	
	1 (1–2)	1 (1–3)	
**Area of detachment**, clock hours Mean ± SD Median (min–max)			0.166 ^†^
	4.2 ± 1.7	3.0 ± 1.4	
	4 (2–7)	3 (1–5)	
**Macular Status**, *n* (%) Attached Detached			0.653 *
	9 (52.9)	5 (71.4)	
	8 (47.1)	2 (28.6)	
**PVR Grade**, *n* (%) Grade A Grade B			0.552 *
	15 (88.2)	5 (71.4)	
	2 (11.8)	2 (28.6)	

Min-max, minimum–maximum; PVR, proliferative vitreoretinopathy; SD, standard deviation; * Fisher’s Exact Test; ^†^ Mann–Whitney U Test.

## Data Availability

The data of the study are available from the corresponding author upon request.

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
