# Peer review of "A Useful Method for the Practice of Pneumatic Retinopexy: Slit-Lamp Laser Photocoagulation through the Gas Bubble"

_jpm, 2023, doi:10.3390/jpm13050741_

Round 1

Reviewer 1 Report

PR is a well-known procedure for RRD. The reviewer wonders what is new in this study.

Author Response

PR is a well-known procedure for RRD. The reviewer wonders what is new in this study.

Thank you for your comment. As you and we also mentioned in the article, pneumatic retinopexy is a 100-year-old technique. Although the indications have changed somewhat over time, it can be assumed that there is a consensus today. However, we see that even in a randomized controlled trial like PIVOT, still no single method is used in the retinopexy phase. In addition, these studies say nothing about why indirect ophthalmoscopy is preferred. Apart from that, we have discussed that an indirect ophthalmoscope may be insufficient in scleral buckling surgery where imaging is very important. We thought it might be useful if we extrapolated this information to the retinopexy phase. Although, as you said, our study was not technically innovative, this discussion of the retinopexy method was a situation that we have never come across in literature searches. From this point of view, we can say that this is the novelty of our study. Finally, we believe that for surgeons who have little experience with indirect Ophthalmoscopes and avoid performing pneumatic retinopexy, a study performed only with a slit lamp, which they are more familiar with in daily use, will give surgeons confidence to achieve results close to the literature.

Reviewer 2 Report

Dear authors,

The study your study about a method (gas bubble under a slit-lamp biomicroscope using a wide-field contact lens) to treat RRD with pneumatic retinopexy is interesting, but overall the article needs to be improved. 

The introduction is poor and small, the RRD must be introduced. At the beginner, I suggest a small introduction about RRD and an explanation about why this eye condition can develop. This can help the readers to better understand you study.

Material and Methods: It's really confusing, I suggest a table to better divide the patients and the procedures used.

Results: few patients (just a result in this session) and it is not described or mentioned any limitation or complications about the methodology.

Author Response

Dear authors,

The study your study about a method (gas bubble under a slit-lamp biomicroscope using a wide-field contact lens) to treat RRD with pneumatic retinopexy is interesting, but overall the article needs to be improved. 

The introduction is poor and small, the RRD must be introduced. At the beginner, I suggest a small introduction about RRD and an explanation about why this eye condition can develop. This can help the readers to better understand you study.

Thank you for your comment. Introduction expanded by adding this paragraph

‘Since its inception, pars plana vitrectomy (PPV) has become the most preferred surgical method for treating RRDs in the world [8-10]. A prospective, randomized-controlled, multicenter study comparing SB and PPV reported a single-procedure success rate of 63.6% and 63.8% and a final anatomical success rate of 96.6% and 96.7% for SB and PPV, respectively[11]. PR and PPV for RRDs were recently compared in a prospective, randomized-controlled, multicenter study[12]. The single-procedure success rates at 12 months were 80.8% and 93.2%, and final anatomical success rates were 98.7% and 98.6% for PR and PPV, respectively. One of the limitations of this study was the inhomogeneous retinopexy method used in the PR group, including cryotherapy alone, laser retinopexy alone, or a combination of both. Therefore, the effects of the retinopexy techniques could not be evaluated.[12]

Material and Methods: It's really confusing, I suggest a table to better divide the patients and the procedures used.

Thank you for your comment, However, since the same procedure is used for all patients, we could not understand what type of table you would like

Results: few patients (just a result in this session) and it is not described or mentioned any limitations or complications about the methodology.

In the last paragraph, we mentioned ‘the number of eyes involved in the study is relatively small compared to other PR studies’ as a main limitation of our study. We did not experience any complications with this method.

Reviewer 3 Report

The authors report the results of the use of pneumatic retinopexy with slit-lamp laser photocoagulation through the gas bubble in a small sample.

The study has the peculiarity of using a precise method of retinopexy (slit-lamp laser photocoagulation through the gas bubble) in contrast to other reports that include indistinctively laser performed with slit-lamp o binocular ophthalmoscope, or cryotherapy to induce chorio-retinal adhesion.

The study did not reveal any innovative information on the subject and the sample is quite small.

 Several issues need to be clarified:

Page 2 line 71 The authors say: “The patients were hospitalized overnight after the injection in the face-down position without a steamroller maneuver and reevaluated within 24 hours….

When was the patient positioned to allow the bubble to cover the retinal break?

It was always the same procedure? Is not uncommon to have more than one bubble that make very difficult to apply the laser through the gas bubble.

Also, in the case of breaks near to 9 or 3 hours, it is easier to apply the laser out of the gas bubble

Page 3 line 103. When the authors say “symptom onset”… Are they speaking about posterior detachment symptoms (photopsias/miodesopsias) or retinal detachment symptoms (loss of visual field or visual acuity loss), or both? It is a bit surprising that most patients presented within ten days of symptom onset and most of them had macula on retinal detachment. 

Page 3 line 106: the authors say “18 (75.0%) of the eyes were phakic”. Nevertheless in table 1 there were 6 phakic patients and 18 pseudophakic

References. Please follow the recommendation of the journal:

Author 1, A.B.; Author 2, C.D. Title of the article. Abbreviated Journal Name Year, Volume, page range.

Please change the “:” after the names of the authors for ”.” following the recommendation of the journal. 

Also change the full name of the journal for the abbreviated journal name 

Page 6 line 246

The year of publication is 1911 (not 2005)

Page 6 line 248: The correct reference is 

Dominguez A: Cirugía precoz y ambulatoria del desprendimiento de retina. Arch Soc Esp Oftalmol 1985, 48:47-54.

Page 7 line 298: reference 25. The name of the journal is missing. The correct reference is:

Hong IH, Jeon GS, Han JR. Comparison of Scleral Buckling and Vitrectomy Using Wide Angle Viewing System for Rhegmatogenous Retinal Detachment. Semin Ophthalmol. 2020 17;35(5-6):307-312. 

Page 7 line 306: reference 29. The name of the journal is missing. The correct reference is:

De Zanet S, Rudolph T, Richa R, Tappeiner C, Sznitman R. Retinal slit lamp video mosaicking. Int J Comput Assist Radiol Surg. 2016,11:1035-41.

Author Response

The authors report the results of the use of pneumatic retinopexy with slit-lamp laser photocoagulation through the gas bubble in a small sample.

The study has the peculiarity of using a precise method of retinopexy (slit-lamp laser photocoagulation through the gas bubble) in contrast to other reports that include indistinctively laser performed with slit-lamp o binocular ophthalmoscope, or cryotherapy to induce chorio-retinal adhesion.

The study did not reveal any innovative information on the subject and the sample is quite small.

Thank you for your comment. As you and we also mentioned in the article, pneumatic retinopexy is a 100 year old technique. Although the indications have changed somewhat over time, it can be assumed that there is a consensus today. However, we see that even in a randomized controlled trial like PIVOT, still no single method is used in the retinopexy phase. In addition, these studies say nothing about why indirect ophthalmoscopy is preferred. Apart from that, we have discussed that an indirect ophthalmoscope may be insufficient in scleral buckling  surgery where imaging is very important. We thought it might be useful if we extrapolated this information to the retinopexy phase. Although, as you said, our study was not technically innovative, this discussion of the retinopexy method was a situation that we have never come across in literature searches. From this point of view we can say that this is the novelty of our study. Finally, we believe that for surgeons who have little experience with indirect ophthalmoscopes and avoid performing pneumatic retinopexy, a study performed only with a slit lamp, which they are more familiar with in daily use, will give surgeons confidence to achieve results close to the literature. For your second comment, the number of patients in our study is low compared to similar studies, we mentioned this as a limitation.

 Several issues need to be clarified:

Page 2 line 71 The authors say: “The patients were hospitalized overnight after the injection in the face-down position without a steamroller maneuver and reevaluated within 24 hours….

When was the patient positioned to allow the bubble to cover the retinal break?

Thank you for your question. From the 1st day of post injection (starting from first evaluation) the patients took an upright position to allow the bubble to cover the retinal break

It was always the same procedure? Is not uncommon to have more than one bubble that make very difficult to apply the laser through the gas bubble.

One of the most common complications of gas injection is fish egg formation. It is known that slow and controlled injection through the gas buble will aid the formation of a single bubble. This method was used in our series and there was no complication in this regard.

Also, in the case of breaks near to 9 or 3 hours, it is easier to apply the laser out of the gas bubble

As you mentioned, in tears close to 9 and 3 hours, the tear is usually out of the gas and laser can be performed around it. Since our series consisted of a homogeneous group in which laser was applied only through gas, these patients were not included. As stated in the method section, the tears to be completely covered by the gas between 8 and 4 hours were taken.

Page 3 line 103. When the authors say “symptom onset”… Are they speaking about posterior detachment symptoms (photopsias/miodesopsias) or retinal detachment symptoms (loss of visual field or visual acuity loss), or both? It is a bit surprising that most patients presented within ten days of symptom onset and most of them had macula on retinal detachment. 

Most patients (79.2%) presented within ten days of symptom onset with a mean symptom duration of 8.7±9.2 (6.0; 1-30) days changed as Most patients (79.2%) presented within ten days of visual loss or visual field loss  with a mean symptom duration of 8.7±9.2 (6.0; 1-30) days.

When we searched the literature for your comment on symptom onset of macular detachments, we found that in different studies, macula on detachments can occur up to 60 days of symptom onset.

In this study, which included 27 patients with macular on detachment, 19 patients presented with prominent floaters or visual field defects within 10 days. 2 patients had macular on detachment and presented within 60 days.

Akiyama K, Fujinami K, Watanabe K, Noda T, Miyake Y, Tsunoda K. Macular dysfunction in patients with macula-on rhegmatogenous retinal detachments. Br J Ophthalmol. 2019;103(3):404-409. doi:10.1136/bjophthalmol-2018-312153

In another study comparing macular on and off macular detachments, the mean time from symptom onset to surgery was 7 days in macula on group.

Rezar S, Sacu S, Blum R, Eibenberger K, Schmidt-Erfurth U, Georgopoulos M. Macula-On Versus Macula-Off Pseudophakic Rhegmatogenous Retinal Detachment Following Primary 23-Gauge Vitrectomy Plus Endotamponade. Curr Eye Res. 2016;41(4):543-550. doi:10.3109/02713683.2015.1031351

For this reason, we think that it is possible to expect macula on detachment in patients who apply within an average of 10 days.

Page 3 line 106: the authors say “18 (75.0%) of the eyes were phakic”. Nevertheless in table 1 there were 6 phakic patients and 18 pseudophakic

Thanks for your caution. The correct information is as in the text, it is "18 (75.0%) of the eyes were phakic". The table has been corrected.

References. Please follow the recommendation of the journal:

Author 1, A.B.; Author 2, C.D. Title of the article. Abbreviated Journal Name Year, Volume, page range.

Please change the “:” after the names of the authors for ”.” following the recommendation of the journal. 

All “:” after the names of the authors for changed to ”.”

Also change the full name of the journal for the abbreviated journal name 

Page 6 line 246

The year of publication is 1911 (not 2005)

Ohm DJ: Über die Behandlung der Netzhautablösung durch operative Entleerung der subretinalen Flüssigkeit und Einspritzung von Luft in den Glaskörper. Albrecht von Graefes Archiv für Ophthalmologie 2005, 79:442-450. changed as suggested

Ohm DJ: Über die Behandlung der Netzhautablösung durch operative Entleerung der subretinalen Flüssigkeit und Einspritzung von Luft in den Glaskörper. Albrecht von Graefes Archiv für Ophthalmologie 1911, 79:442-450.

Page 6 line 248: The correct reference is 

Dominguez A: Cirugía precoz y ambulatoria del desprendimiento de retina. Arch Soc Esp Oftalmol 1985, 48:47-54.

A D: Cirugía precoz y ambulatoria del desprendimiento de retina. Arch Soc Esp Oftalmo 1985, 48:47-54. changed as suggested

Dominguez A: Cirugía precoz y ambulatoria del desprendimiento de retina. Arch Soc Esp Oftalmol 1985, 48:47-54.

Page 7 line 298: reference 25. The name of the journal is missing. The correct reference is:

Hong IH, Jeon GS, Han JR. Comparison of Scleral Buckling and Vitrectomy Using Wide Angle Viewing System for Rhegmatogenous Retinal Detachment. Semin Ophthalmol. 2020 17;35(5-6):307-312.  

Hong IH, Jeon GS, Han JR: Comparison of Scleral Buckling and Vitrectomy Using Wide Angle Viewing System for Rhegmatogenous Retinal Detachment. 2020, 35(5-6):307-312. changed as suggested

Hong IH, Jeon GS, Han JR. Comparison of Scleral Buckling and Vitrectomy Using Wide Angle Viewing System for Rhegmatogenous Retinal Detachment. Semin Ophthalmol. 2020 17;35(5-6):307-312.  

Page 7 line 306: reference 29. The name of the journal is missing. The correct reference is:

De Zanet S, Rudolph T, Richa R, Tappeiner C, Sznitman R. Retinal slit lamp video mosaicking. Int J Comput Assist Radiol Surg. 2016,11:1035-41.

De Zanet S, Rudolph T, Richa R, Tappeiner C, Sznitman R: Retinal slit lamp video mosaicking. 2016, 11(6):1035-1041. changed as suggested De Zanet S, Rudolph T, Richa R, Tappeiner C, Sznitman R. Retinal slit lamp video mosaicking. Int J Comput Assist Radiol Surg. 2016,11:1035-41.

Round 2

Reviewer 1 Report

The success rate is not high, sorry.

Reviewer 2 Report

Dear Authors,

Thanks for your editing.